# Fabrication of Ultraviolet Photodetectors Based on Fe-Doped ZnO Nanorod Structures

**DOI:** 10.3390/s20143861

**Published:** 2020-07-10

**Authors:** Yen-Lin Chu, Sheng-Joue Young, Liang-Wen Ji, I-Tseng Tang, Tung-Te Chu

**Affiliations:** 1Department of Electro-Optical Engineering & Institute of Electro-Optical and Materials Science, National Formosa University, Yunlin 632, Taiwan; 10576123@gm.nfu.edu.tw; 2Department of Electronic Engineering, National Formosa University, Yunlin 632, Taiwan; 3Department of Greenergy, National University of Tainan, Tainan 701, Taiwan; tangitseng@gmail.com; 4Department of Mechanical Automation Engineering, Kao Yuan University, Kaohsiung 821, Taiwan; t30039@cc.kyu.edu.tw

**Keywords:** zinc oxide, ultraviolet, nanorods, Fe, photodetectors

## Abstract

In this paper, 100 nm-thick zinc oxide (ZnO) films were deposited as a seed layer on Corning glass substrates via a radio frequency (RF) magnetron sputtering technique, and vertical well-aligned Fe-doped ZnO (FZO) nanorod (NR) arrays were then grown on the seed layer-coated substrates via a low-temperature solution method. FZO NR arrays were annealed at 600 °C and characterized by using field emission scanning microscopy (FE-SEM) and X-ray diffraction spectrum (XRD) analysis. FZO NRs grew along the preferred (002) orientation with good crystal quality and hexagonal wurtzite structure. The main ultraviolet (UV) peak of 378 nm exhibited a red-shifted phenomenon with Fe-doping by photoluminescence (PL) emission. Furthermore, FZO photodetectors (PDs) based on metal–semiconductor–metal (MSM) structure were successfully manufactured through a photolithography procedure for UV detection. Results revealed that compared with pure ZnO NRs, FZO NRs exhibited a remarkable photosensitivity for UV PD applications and a fast rise/decay time. The sensitivities of prepared pure ZnO and FZO PDs were 43.1, and 471.1 for a 3 V applied bias and 380 nm UV illumination, respectively.

## 1. Introduction

People’s lives have become more comfortable and convenient with the rapid development of the economy, military and industrial technology in the past decade. However, this development has caused several adverse effects, such as greenhouse effect, air pollution and acid rain, which have damaged the ecological balance of Nature. It has also affected the rapidly breaking ozone layer causing a loss of the atmosphere’s ultraviolet (UV) filter function. Excessive UV light can directly damage the human body, leading to severe sunburn, skin aging, and skin cancers (e.g., basal cell carcinoma, squamous cell carcinoma, and malignant melanoma) [1,2,3]. Therefore, high performance UV photodetectors (PDs) have been developed to solve these problems. In recent years, UV PDs have played an important role in different applications which have been extensively explored in the areas of human health, pollution monitoring, flame detection, water sterilization, and missile warning systems [4]. Many metal oxide semiconductors (MOSs) are also employed in UV detection; examples include silicon carbide (SiC), gallium nitride (GaN), titanium dioxide (TiO_2_), and zinc oxide (ZnO) [5,6,7,8]. Among them, ZnO is one of the most explored, partly because of its distinct properties (e.g., non-toxicity, high electron mobility, biocompatibility, and low cost) and potential applications in optoelectronic components. ZnO is an n-type MOS and a group II–VI material, its band gap is about 3.37 eV, and it has a large exciton binding energy (60 meV) at room temperature. Moreover, it has a hexagonal structure, and the properties of lattice constants are a = 3.24–3.26 Å and c = 5.13–5.43 Å [9,10]. ZnO has attracted tremendous and wide interest in different fields, such as field-emission displays (FEDs), PDs, gas sensors, and pH sensors [11,12,13,14]. Therefore, ZnO has gradually become a promising material for various applications in modern times.

Based on these properties, ZnO materials have been extensively employed in UV PDs due to their excellent electronic and photonic properties. ZnO nanostructures are also applied to UV PDs because of their high aspect ratio, high on/off current ratio, rapid response and recovery rate and high photoconductive gain [15]. In previous reports, ZnO nanostructures were synthesized by using various methods, including pulsed laser deposition (PLD), molecular beam epitaxy (MBE), metal organic chemical vapor deposition (MOCVD), and chemical bath deposition (CBD) methods [16,17,18,19]. Among the different growth methods, CBD is an uncomplicated growth method under low temperature conditions and also an inexpensive method. It was employed in the preparation of ZnO nanostructures in our experiments owing to these outstanding advantages. Until now, ZnO UV PDs have been used to fabricate various structures such as Schottky junctions, P-N junctions and metal–semiconductor–metal (MSM) structures [20,21,22]. The performance of these devices usually depends on the preparation of ZnO nanomaterials. Moreover, ZnO exhibits a large adsorption/desorption behavior for oxygen on the surface, which can reduce the dark current and enhance the responsivity of UV PDs. However, the adsorption/desorption process of oxygen on ZnO surface usually leads to a long rise/recovery time of ZnO UV PDs, which limits its practical applications [23]. Iron (Fe) is a good doping candidate to overcome this drawback. It is a donor dopant with good electrical and thermal conductivity, which can improve the responsivity of UV PDs [24].

In this work, we report on the structure morphology, crystal lattice, and optical properties of synthesized Fe-doped ZnO (FZO) nanorod (NR) arrays. These FZO NR arrays are prepared through a simple CBD method with a photolithography procedure. The electrical characteristics of the resulting FZO PD samples were also investigated by several instrumental techniques.

## 2. Experimental

PDs based on FZO NR arrays were fabricated in three stages: (1) in a sputtering system, a ZnO (3-in, 99.9%) target under the base working pressure in the growth chamber was kept at 5 × 10^−6^ Torr, while the applied radio frequency (RF) power was 30 W, the ratio of Ar/O_2_ gas flow was maintained at 12/1 (sccm), and the substrate temperature was 30 °C. The 100 nm-thick ZnO seed layer prepared in this investigation was firstly deposited on a Corning glass substrate (1 cm × 1 cm) via a RF magnetron sputtering technique (13.56 MHz) at room temperature for 80 min. (2) The standard photolithography technique was then applied to define a micro-pattern on the substrate by using a shadow mask. A 100 nm-thick silver (Ag) film used as an electrode was evaporated on the ZnO seed layer using an electron beam evaporation to form the MSM structure. During electrode evaporation, the current and voltage of the electron beam were about 15 mA and kept above 5 kV, respectively. The deposition time was almost 90 min, and temperature was maintained at 110 °C. Subsequently, a lift-off process was used to form a micro-patterned interdigital transducer (IDT) structure. The fingers of the Ag contact electrodes were 10 μm long and 3 μm wide with 3 μm spacing (the active sensing area was 10 μm × 30 μm). (3) Chemical reagents purchased from Sigma-Aldrich (Merck KGaA, Darmstadt, Germany) were analytical grade and used without further purification in this experiment to fabricate FZO NRs. Aqueous solution of 0.03 M zinc nitrate hexahydrate [Zn(NO_3_)_2_·6H_2_O, 99.99%], 0.03 M hexamethylenetetramine (C_6_H_12_N_4_, HMTA), and 0.4 mM ferric nitrate nanohydrate [Fe(NO_3_)_3_·9H_2_O, 98.4%] was mixed with deionized (DI) water and then stirred for 20 min at 60 °C. When the solution was mixed completely, the ZnO seed layer grown on a substrate was immersed in a 100 mL serum bottle containing the mixed aqueous solution for 3 h at 95 °C in an oven. After completion of the reaction, the serum bottle was cooled to room temperature, and the sample was washed twice with DI water and dried in air for 20 min. DI water from a Milli-Q system (18.2 MΩ·cm) was used throughout the experiments. Finally, the samples were annealed in a high vacuum state (~1.5 × 10^−5^) at 600 °C for 10 min to obtain high-crystallinity nanostructures. Before the UV PDs were manufactured, cleaning the substrate was also an important process because it will affect the growth condition of nanostructures. Thus, all Corning glass substrates were cleaned for 10 min with acetone, isopropyl alcohol, and DI water. Details of the manufacture of the FZO PDs are shown in Figure 1.

The surface morphology and lattice properties of the synthesized ZnO NRs with Fe content were observed by using a field emission scanning electron microscope (FE-SEM, JSM-6700F, JEOL, Tokyo, Japan) operated at 15 kV and a high-resolution transmission electron microscope (HR-TEM, JEOL JEM-2100F CS-STEM, Tokyo, Japan) to explore the properties of the material and the device. Fe contents in the samples were checked by using an energy dispersive X-ray (EDX) spectrometer. An X-ray diffractometer (XRD, MO3XHF22 MAC-Science, Kanagawa, Japan) with Cu Kα radiation (λ = 0.15418 nm) was used to explore crystallographic and structural characteristics of the as-grown pure ZnO and FZO NR arrays. Optical properties of the as-prepared samples were investigated by using a photoluminescence spectrometer (PL, Labram HR, Horiba, Ltd., Horiba, Ltd., Kyoto, Japan) with a He–Cd laser (325 nm, 5 mW) as the excitation light source. An HP-4156C semiconductor parameter analyzer (Agilent Technologies Inc, Santa Clara, CA, USA.) was then employed to measure current-time (*I-T*), current-voltage (*I-V*), photocurrent, and dark current characteristics of fabricated pure ZnO and FZO PD samples. The photoresponsivity (*A/W*) measurements were also conducted using a TRIAX 180 monochromator system (Horiba, Ltd., Kyoto, Japan) with a 300 W xenon arc lamp light source and a 2410 semiconductor system (Keithley, Agilent Technologies Inc, Santa Clara, CA, USA.) at room temperature.

## 3. Results and Discussion

The surface morphology and structural dimensions of samples were characterized by using FE-SEM. Figure 2 shows that NR arrays grew on the substrate from the solutions containing equi-molarity concentrations of (0.03 M) Zn(NO_3_)_2_·6H_2_O and (0.03 M) HMTA in the absence of Fe^3+^ ions and with Fe-doping. During the growth of the nanostructure, ZnO film was used as seed layer to assist the growth of NR arrays. Zn^2+^ and OH^−^ ions provided by hydration of Zn(NO_3_)_2_·6H_2_O and HMTA likely reacted with OH^−^ and formed soluble Zn(OH)_2_ complexes, which are the growth units of ZnO structures. Finally, a ZnO nanostructure was generated by the decomposition of Zn(OH)_2_. Moreover, Fe^3+^ ions were mostly in the form of Fe precursors and Fe complexes. Figure 2a–d show the top view and cross-section of pure ZnO and FZO NRs. Growth of the FZO NRs proceeded by the following chemical reaction [25,26]:Zn(NO_3_)_2_·6H_2_O → Zn^2+^ + 2NO_3_^−^ + 6H_2_O(1)
Fe(NO_3_)_3_·9H_2_O → Fe^3+^ + 3NO_3_^−^ + 9H_2_O(2)
C_6_H_12_N_4_ + 6H_2_O → 6HCOH + 4NH_3_(3)
NH_3_ + H_2_O → NH_4_^+^ + OH^−^(4)
Zn^2+^ + 2OH^−^ → Zn(OH)_2_(5)
Fe^3+^ + 3OH^−^ → Fe(OH)_3_(6)
Zn^2+^ + Fe^3+^ + 5OH^−^ → Zn(OH)_2_ + Fe(OH)_3_(7)
(8)$$\mathrm{Zn}(\mathrm{OH}{)}_{2}\;\overset{\Delta }{↔}\;\mathrm{ZnO}+H_{2}O$$
(9)$$2\mathrm{Fe}(\mathrm{OH}{)}_{3}\;\overset{\Delta }{↔}\;{\mathrm{Fe}}_{2}O_{3}+3H_{2}O$$
(10)$$\mathrm{Zn}(\mathrm{OH})2_{\;}+2\mathrm{Fe}(\mathrm{OH}{)}_{3}\;\overset{\Delta }{↔}\;\mathrm{ZnO}+{\mathrm{Fe}}_{2}O_{3}+4H_{2}O$$

The figure shows the formation of highly oriented hexagonal-shaped NRs nearly vertical to the substrate surface. In pure ZnO FE-SEM, top view and cross-section images exhibit that the diameter and length of the NRs were about 97 nm and 1.3 μm, respectively. After Fe-doping, the diameter and length of NRs were increased slightly. The results of this paper are similar to those of a previous report by Liu et al. and Sahai et al [26,27].

XRD measurements were carried out to determine the crystallinity properties of pure ZnO and FZO NRs shown in Figure 3. The XRD diffraction peaks of all synthesized samples clearly reflect a wurtzite structure with typical hexagonal crystals (JCPDS Card No. 36-1451). The XRD pattern of pure ZnO reveals three peaks at 2θ = 34.54°, 47.69°, and 62.94°, which corresponded to the (002), (102), and (103) planes. The XRD pattern of the Fe-doped sample is similar to that of pure ZnO structure. No reflection peaks of Fe_2_O_3_ (JCPDS Card No. 33-0664) or Fe_3_O_4_ (JCPDS Card No. 85-1436) can be seen in the diffraction patterns. The absence of impurity peaks suggests that the samples are highly crystalline and have good crystal quality [28]. A strong (002) preferential orientation is also observed in pure ZnO and FZO structure. This finding implies that all samples are single crystalline and vertically grown on the substrate. Moreover, based on Bragg’s Law formula, the shift position in the diffraction angles at (002) peak is derived from the decrease of d-spacing [29]:n λ = 2d sin θ(11)
where n, λ, θ, and d are the order of diffraction, wavelength of the x-rays, angle of diffraction, and distance between planes, respectively. The inset of Figure 3 shows that the location of the (002) diffraction peak slightly shifts toward higher diffraction angles, which indicates the change in d-spacing. This result implies that the ionic radii value of Fe^3+^ (0.068 nm) is smaller than that of Zn^2+^ ion (0.074 nm), and Fe atoms may substitute Zn^2+^ sites or incorporate interstitially in the ZnO structure [30,31].

The optical properties of pure ZnO and FZO PDs were investigated via room temperature PL measurement. PL spectroscopy is a sensitive, non-destructive technique and is appropriate for identifying intrinsic and extrinsic defects in materials. Figure 4 shows the normalized PL spectra of pure ZnO and FZO NRs with a He–Cd laser (325 nm, 5 mW) as the excitation-light source. According to a previous report [32], the PL spectrum of pure ZnO structure consists of two major peak regions: (1) a sharp peak located at UV emission (near-band emission, NBE) and (2) a broad peak in the visible green emission (deep-level emission, DLE). The appearance of UV emission and visible green emission reveals the presence of native, intrinsic defects in ZnO, such as interstitial zinc (Zn_i_), zinc vacancy (V_Zn_), oxygen vacancy (V_O_), interstitial oxygen (O_i_), zinc antisite, and oxygen antisite. PL spectra of FZO NRs in the wavelength range of 300–700 nm are also observed. All spectra on the strong UV peak vary from 379 nm to 385 nm for 0 to 0.4 mM Fe-doping. Compared with the PL result of pure ZnO, it was revealed that it was red-shifted in the UV emission region and the visible green emission of the FZO NRs decreased slightly. These phenomena can be explained as follows: (1) band gap varied due to sp–d exchange interactions in the ZnO matrix; and (2) Fe-doping may supply competitive pathways for recombination, resulting in quenching of the visible green emission. These phenomena are similar to the result reported earlier for Fe-doping in ZnO structure [33,34,35].

TEM, HR-TEM, SAED, and EDX images of the synthesised FZO NRs were observed in Figure 5. Figure 5a shows a low-magnification TEM image of the single FZO NR structure. Figure 5b reveals that the d-spacing between adjacent lattice planes was 0.258 nm, which appeared to be oriented in the c-axis direction. The inset of Figure 5b exhibits that the FZO NR has a single crystalline structure.

Meanwhile, Figure 5d–g show the elemental mapping images of the ZnO NRs with F content. It can be found that Zn (red), O (yellow), and Fe (cyan) elements are uniformly distributed in the FZO NR structure, this means Fe ions may enter into ZnO lattice. The EDX image taken from top view in Figure 2 consists of Zn, O, and Fe peaks and confirms the presence of Fe content about 0.51 at% into the sample, as shown in Figure 5h.

Figure 6 shows *I-T* and *I-V* characteristics of pure ZnO and FZO structure-based UV PDs. Figure 6a,b depict the photocurrent rise and decay obtained by turning the continuous UV illumination on and off with a 3 V applied bias. In this work, rise times (r_1_, r_2_) reveal the time for photocurrent rise to 50% and 90% of the peak value, respectively; and decay times (d_1_, d_2_) indicate the time for the photocurrent decays to 50% and 10% of the peak value, respectively. In the coefficients of all samples, the transient’s response during turn-on and turn-off process can be intently fitted via exponential curves, as indicated in the following equations [36]:(12)$$\mathrm{Turn}-\mathrm{on}:\;I(t)=I_{0}(t)\;\left[1-\exp{\left(\frac{-t}{\tau_{r}}\right)}^{\beta }\right]$$
(13)$$\mathrm{Turn}-\mathrm{off}:\;I(t)=I_{0}(t)\;\left[\exp{\left(\frac{-t}{\tau_{d}}\right)}^{\beta }\right]$$
where I_0_(t), t, β, and τ are the transient current, time after turn-on/turn-off, decay exponent, and time constant, respectively. The fitted exponential curves indicate that β is about ~1. Compared with pure ZnO NRs, the rise times (r_1_,r_2_) of FZO were 2 and 46 s, respectively; and the decay times (d_1_,d_2_) were 2 and 37 s, respectively. As a result, both times of FZO PD samples are faster than those of pure ZnO PD samples. The measurement of continuity is shown in the insets of Figure 6a,b. They show that the reproducibility of pure ZnO and FZO samples is stable (the cyan block area is under 380 nm UV irradiation). The carrier relaxation phenomenon can be divided into two electronic processes: first, electron loss due to recombination at the deep defect states; second, electron trapping by the surface states [36,37]. *I-V* curves for the UV sensing of ZnO PDs with and without Fe content are shown in Figure 6c,d. The result displays all measured PDs from −5 V to +5 V bias voltages. 

These PD devices are prepared as MSM structures that belong to the back-to-back Schottky junction. This designed structure is similar to that reported earlier for the FZO structure [38]. As a result, with a 3 V applied bias and 380 nm UV illumination, the sensitivities (I_ph_/I_dark_) of fabricated pure ZnO and FZO PDs were 43.1, and 471.1, respectively. Thus, Fe-doping can reduce the dark current and enhance the electrical properties of PDs, as shown in Table 1.

The spectral responsivities of pure ZnO and FZO PDs with an applied bias of 3 V and under 380 nm illumination are shown in Figure 7. Responsivities are nearly constant in the UV region (300 to 375 nm). Here, we define UV-to-visible rejection ratio as the responsivity measured at 370 nm divided by the responsivity at 550 nm. The UV-to-visible rejection ratios for pure ZnO and FZO MSM PDs were 96.03 and 1273.61, respectively. These results show that the responsivities for the prepared pure ZnO and FZO PDs were 0.298 and 0.758 A/W, respectively. A calibrated measurement of the monochromated xenon arc lamp irradiance was about 30 W/m^2^. The responsivity (R) of a PD is defined as the following equation [39]:R = I_ph_/P_inc_(14)
where I_ph_ and P_inc_ are the photocurrent and incident optical power, respectively. As a donor, Fe^3+^ ion usually provides a pair of electrons in the conduction band, thus improving the photoconductivity [40]. As a result, FZO PDs have better performance than pure ZnO PDs. The large UV-to-visible rejection ratio of FZO samples also implies that the PD with Fe content is potentially useful for practical applications.

Figure 8 shows a model mechanism for charge transport in FZO NR-based MSM PDs. Oxygen (O_2_) molecules are adsorbed onto the surface of ZnO material as negatively charged ions via capturing free electrons from the n-type ZnO structure, which can produce a low conductivity depletion layer near the surface of pure ZnO NRs in dark conditions [O_2_(gas) + e^−^ → O_2_^−^(adsorption)]. However, under UV irradiation, UV light (hv > Eg) photogenerates electron-hole pairs and discharges the adsorbed oxygen (O_2_^−^) ions through surface electron-hole recombination on pure ZnO NRs [hv → e^−^ + h^+^]. The electron of the pair stays in the conduction band, thereby raising conductivity and decreasing resistivity [O_2_^−^(adsorption) + h^+^ → O_2_(gas)]. By contrast, FZO NR arrays with high surface volume ratio can lead to their adsorption onto and desorption from the surface of FZO [41,42]. Therefore, their response speed and the carrier concentration in the FZO structure can be effectively improved; they can further contribute to the formation of more oxygen vacancies, thus enhancing the photocurrent of FZO PDs. Zn^2+^ ions in ZnO lattice are easily replaced by Fe^3+^ ions. However, the positive charge of substituted Zn^2+^ site (formation of electron-donor defects [Fe_Zn_]) must be compensated by releasing electrons to keep its electrical neutrality. After Fe-doping, electrons are also introduced into the conduction band of the sample, which enhances the concentration of free electrons, resulting in the resistivity decrease of the FZO material, that is, the responsivity increases. The above description can be written as the following equation [43,44]:(15)$${\mathrm{Fe}}_{2}O_{3(s)}\;\overset{\mathrm{ZnO}}{\to }\;2{\mathrm{Fe}}_{\mathrm{Zn}}+3O_{o}^{X}+2e^{-}$$
where in the Kroger–Vink notation, Fe_Zn_ is an Fe ion sitting on a Zn lattice site with one positive charge, and $O_{o}^{X}$ is an oxygen ion sitting on an O lattice site with a neutral charge. Based on the trapping mechanism related to oxygen adsorption/desorption, FZO NRs are remarkable materials for UV detection, as shown in Figure 8a.

The schematic energy band diagram of the Schottky barrier formed at the Ag/FZO interface in the dark and UV illumination is shown in Figure 8b. To the best of our knowledge, the presence of surface states at the semiconductor and metal interface can affect the Schottky barrier. In this investigation, interface surface states between FZO and Ag, and oxygen adsorption and desorption effect can evidently change the height of the Schottky barrier. In a general dark environment, they form a high-energy barrier in the Schottky behavior (Φ_B_ > Φ_B′_). Under UV irradiation, an electron-hole pair can be effectively produced, and carrier density can be substantially increased, which affects Fermi energy [45,46]. When UV light irradiates on the FZO PDs, it can produce numerous carriers, which can easily exceed the Schottky barrier, making the photocurrent and UV to-visible rejection ratio increase larger than that of pure ZnO PDs. Therefore, responsivities of FZO PDs are improved by the hole-trapping mechanism through oxygen adsorption and desorption in FZO NR arrays, which raises the density of trap states. In addition, this phenomenon can increase in carrier injection and transport, generating a consistent photocurrent. A comparison between the photosensitivity performance of the FZO NRs and these MOS structures previously reported is summarized in Table 2. As can be seen, the PD made of FZO NRs in the present work displays a higher photosensitivity than those reported in other studies, which confirms that FZO structures have a clear advantage over the others for UV sensing.

## 4. Conclusions

In this paper, high single-crystalline FZO NR arrays were successfully synthesized on substrates by using a CBD method. Grown FZO NRs were used to fabricate high-performance UV PDs. FE-SEM and TEM images display the one-dimensional morphology of the FZO NR arrays with a hexagonal wurtzite structure. FZO NRs are structurally uniform and highly oriented preferentially in the c-axis direction with good crystal quality. XRD spectral analysis revealed that the (002) peak position was shifted to a higher angle by Fe-doping in the growth solution. Fe^3+^ ions substituted Zn^2+^ sites and entered into the ZnO lattice with no secondary phases or impurities of Fe_2_O_3_ and Fe_3_O_4_ exhibited in the grown samples. After Fe-doping, PL spectra of the samples revealed red-shifted in the presence of Fe in the ZnO structure. Compared with pure ZnO PDs, FZO NR PDs with 380 nm UV light at 3 V applied bias exhibited a high photocurrent with a sensitivity of 471.1, which is attributed to the large number of electrons. Thus, the rise time and decay time of FZO PD samples are faster than those of pure ZnO PD samples. ZnO MSM PDs prepared with Fe treatment have potential for use in high-performance UV PDs and can be combined in Internet of Things sensor systems to detect high-radiation UV light exposure in the environment.

## Figures and Tables

**Figure 1 sensors-20-03861-f001:**
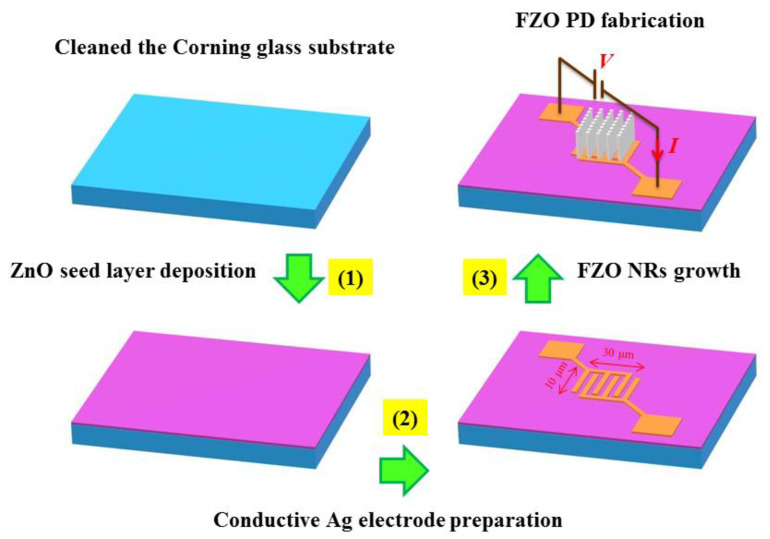
Schematic illustration of fabrication steps of FZO NR-based MSM UV PDs.

**Figure 2 sensors-20-03861-f002:**
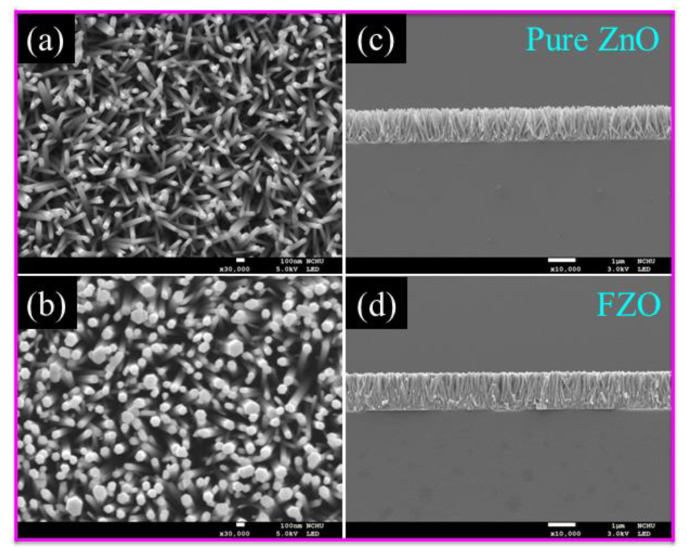
FE-SEM (**a**,**b**) top view and (**c**,**d**) cross-sectional micrographs of ZnO NR arrays prepared via the aqueous solution method with and without Fe content.

**Figure 3 sensors-20-03861-f003:**
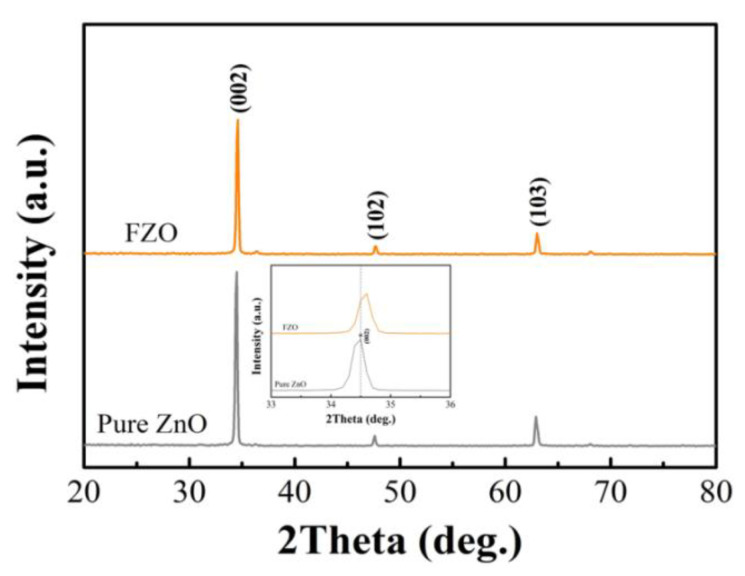
Comparative XRD analysis on pure ZnO and FZO NRs synthesized using the CBD method. [Figure inset shows that the (002) peak of FZO NRs has an angle shift of 0.16° toward a higher angle compared with pure ZnO NRs].

**Figure 4 sensors-20-03861-f004:**
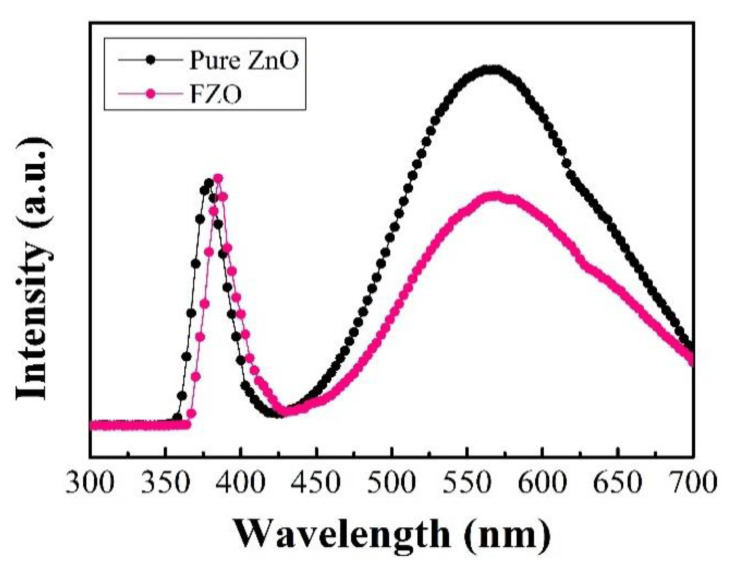
PL spectra of pure ZnO NRs and FZO NRs at 5 W with an excitation source of 325 nm; UV emission of FZO NRs has a red-shifted of 6 nm compared with that of pure ZnO NRs.

**Figure 5 sensors-20-03861-f005:**
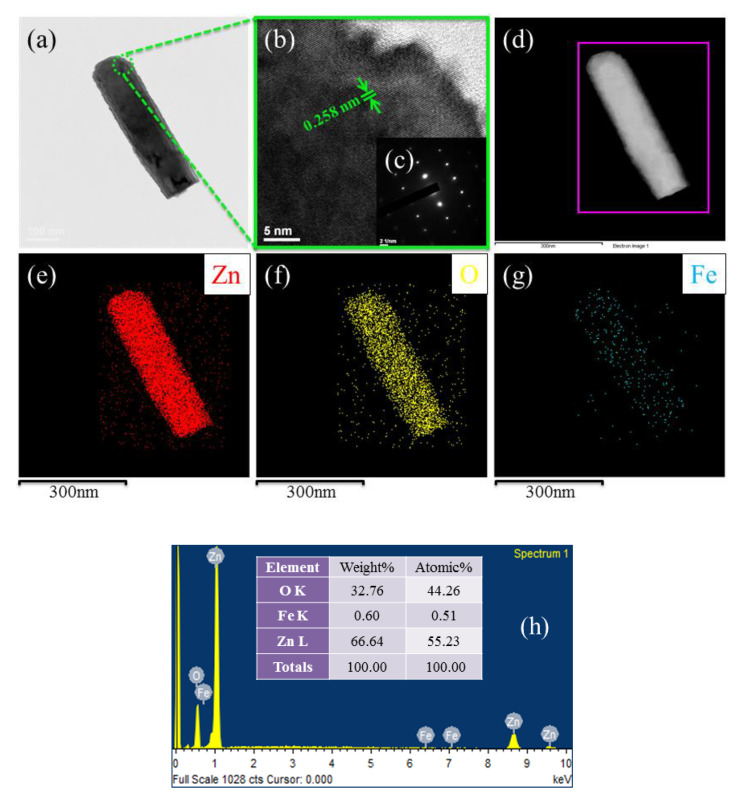
(**a**) Low-magnification BF TEM micrograph of FZO NR sample grown on substrate. (**b**) HR-TEM image of tip of the FZO NRs, where inset (**c**) corresponds to the SAED pattern of FZO NRs. (**d**–**g**) TEM elemental mapping images of as-prepared FZO NR structure for the selected region, displaying the existence of Zn (red), O (yellow), and Fe (cyan). (**h**) EDX spectra of as-synthesized FZO NR arrays.

**Figure 6 sensors-20-03861-f006:**
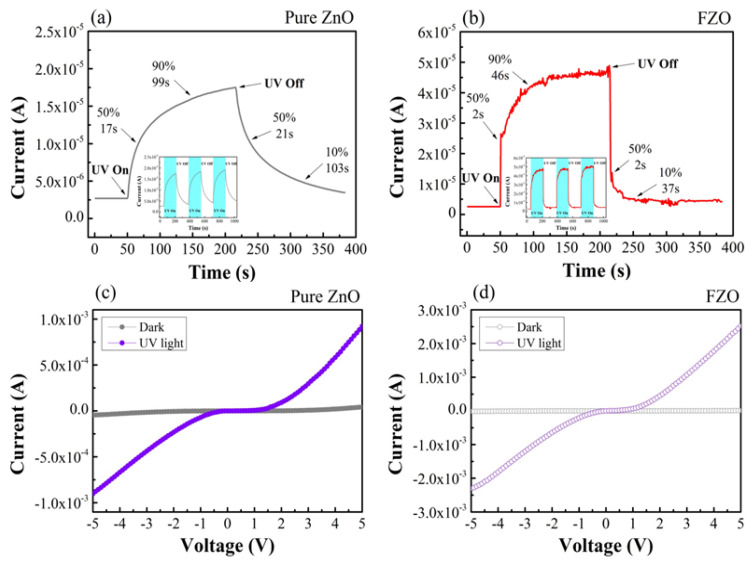
(**a**,**b**) *I*–*T* responses of photocurrent from pure ZnO and FZO NR PDs to UV light at 3 V applied bias. (**c**,**d**) *I*–*V* characteristics of fabricated ZnO MSM PDs with and without Fe content measured in the dark and 380 nm UV light environment.

**Figure 7 sensors-20-03861-f007:**
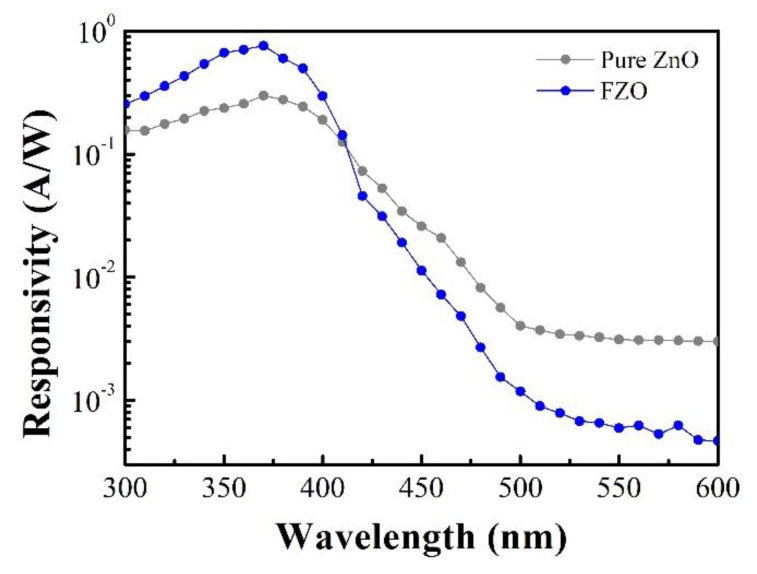
Measured spectral responsivities of MSM PDs based on pure ZnO and FZO NRs with an applied voltage of 3 V at room temperature.

**Figure 8 sensors-20-03861-f008:**
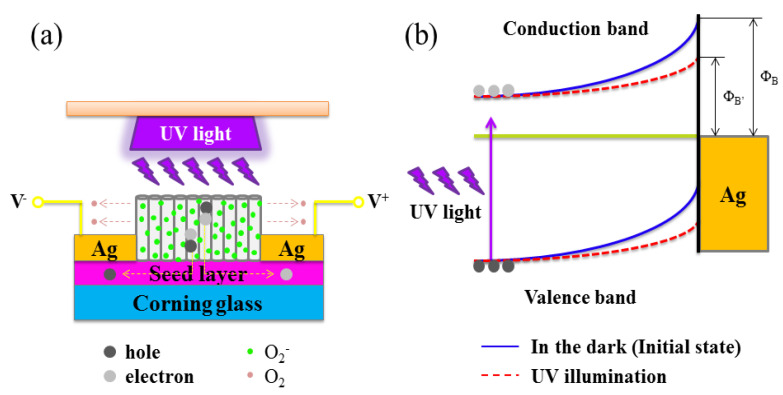
(**a**) Schematic diagram of FZO NR PDs under UV illumination. (**b**) The Schottky barrier formed at the interface between the FZO NRs and Ag electrodes, and the electron-hole pairs were photogenerated under UV light irradiation.

**Table 1 sensors-20-03861-t001:** The results exhibited the parameters of the prepared PD samples with and without Fe content at 3 V.

PD Samples	τ_r1_ (s)	τ_r2_ (s)	τ_d1_ (s)	τ_d2_ (s)	I_dark_ (A)	I_photo_ (A)	Sensitivity
**Pure ZnO**	17	99	21	103	6.89 × 10^−6^	2.97 × 10^−4^	43.1
**FZO**	2	46	2	37	2.25 × 10^−6^	1.06 × 10^−3^	471.1

**Table 2 sensors-20-03861-t002:** Comparison of the photosensitivity characteristics of FZO NRs and other reported MOS structures.

MOS Structures	Bias (V)	Sensitivity (I_ph_/I_d_)	Rise Time (s)	Decay Time (s)	Responsivity (A/W)	Ref.
**Ni-ZnO NRs**	3	393.04	70	43	2.10	[12]
**Ga-ZnO NRs**	1	11.7	29.75	89.67	<0.05	[47]
**Pure ZnO NRs**	1	<10	42.9	132.9	×	[48]
**Fe-ZnO NRs**	1	~49	21.2	24.7	0.06	[48]
**2 at% Cd-ZnO film**	5	93.78	37.03	221.93	×	[49]
**15 at% Mg-ZnO film**	5	71.68	381	122.5	×	[50]
**FZO NRs**	3	471.1	46	37	0.758	Present

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
