# Peer review of "Fabrication of Ultraviolet Photodetectors Based on Fe-Doped ZnO Nanorod Structures"

_sensors, 2020, doi:10.3390/s20143861_

Round 1

Reviewer 1 Report

see attached file

Author Response

Reply to the comments of the editor and reviewer

Journal: Sensors (ISSN 1424-8220)

Manuscript Number: Sensors-853649

Title: Fabrication of Ultraviolet Photodetectors Based on Fe-Doped ZnO Nanorod Structures

=============================================================

Reviewer(s)' Comments to Author:

Reviewer: 1

The paper presents results on the fabrication of Fe-doped ZnO nanorods for the detection of UV light. I will list some major points that need to be addressed before publishing.

  • The structure of the paper and the type of measurements are mostly copied from other works of the same group where they use different metals to dope ZnO nanorods. In particular: a) Cited. “Fabrication and Characterization of Ni-Doped ZnO Nanorod Arrays for UV Photodetector Application”, Yen-Lin Chu et al 2020 J. Electrochem. Soc. 167 067506. b) NOT cited. “Fabrication and Characterization of UV Photodetectors with Cu-Doped ZnO Nanorod Arrays”, Yen-Lin Chu et al 2020 J. Electrochem. Soc. 167 027522. c) Cited. “Fabrication and characterization of homostructured photodiodes with Li-doped ZnO nanorods”, Huang, C., Chu, Y., Ji, L. et al. Microsyst Technol (2020). https://doi.org/10.1007/s00542-020-04854-1.

I approve the concept of publishing more papers instead of one, single big-paper (to increase the number of publications and therefore the h-index), but I do not approve the lack of comparison between the performance of the various detectors. I think this paper is missing an entire section where the performance of this detector is compared with the other metal doped ZnO nanorods. Is iron better than copper, nichel or lithium?

Answer:

Thanks for your kind suggestion!

  • We have perused the reviewers’ comments and suggestions, and made appropriate revisions in the revised manuscript to improve the clarity and delivery of the technical data. We also appreciate the reviewer for giving many constructive comments. According to previous reports, different dopants show different electro-optical characteristics and material properties. We have compared with the previous published articles, there is no repetition problem. Meanwhile, by comparing with previous studies, we have added the table 2 in the revised manuscript (yellow color, line 269–273, 453–454, and 495–496), as seen in table 2 and reference 47-50.

  • The authors should add the following citation: “Noise Properties of Fe-ZnO Nanorod Ultraviolet Photodetectors” S. Chang et al., IEEE Photonics Technology Letters, vol. 25, no. 21, pp. 2089-2092, Nov. 1, 2013. This paper shows that the author Sheng-Joue Young was already working on Fe-doped nanorods back in 2013, so the idea of growing Fe-doped ZnO nanorods for UV detection is not new at all. Even the solution used to dope the nanorods is identical. The authors should explicitly describe what kind of improvements on nanorod growth condition and doping have been made since that experiment, and if these improvements results in a better UV detection.

Answer:

Thanks for your kind suggestion!

  • We have added the reference in the revised manuscript, as seen in reference 48. (yellow color, line 412–413). Compared with a previous report, the growth temperature conditions of nanorods were slightly increased in our experiment. Thus, the nanostructures were obtained in a short time. Additionally, the electrode is miniaturized without large size spacing; the results show the stabilities of pure ZnO and FZO PD samples are better.

  • Line 83: “traditional yellow photolithography” The word “traditional” is not appropriate to describe photolithography. Perhaps you can say “standard”. Also the word “yellow” is not correct, because yellow photolithography does not exist. Maybe the lights in the room are yellow to avoid accidental UV exposure of the sample, but the UV light used in photolithography is not visible (maybe just some deep blue/purple color).

Answer:

Thanks for your kind suggestion!

  • The standard photolithography technique was then prepared and defined a micro-pattern on the substrate by the shadow mask. We have modified the sentences in the revised manuscript (yellow color, line 82–84).

  • Line 87. The deposition time for 100 nm silver layer is extraordinarily long, with a deposition rate of approx 0.2 A/s. Is there a specific reason to use such a small deposition rate?

Answer:

Thanks for your kind suggestion!

  • As we know, the lower deposition rate of silver electrode may deposit completely on the substrate, which can obtain good electrode quality.

  • Line 99. “the samples were annealed in a vacuum state”. Can you be more specific? Was it low vacuum (a few mbar pressure), medium vacuum or high-vacuum?

Answer:

Thanks for your kind suggestion!

  • The samples were annealed in a high vacuum state (~1.5 × 105) at 600 °C for 10 min to obtain high-crystallinity nanostructures. We have modified the sentences in the revised manuscript (yellow color, line 99–100).

  • Line 116. You should add “monochromator” next to the TRIAX 180, because it is not obvious to everyone that the xenon lamp has a monochromator.

Answer:

Thanks for your kind suggestion!

  • The photoresponsivity (A/W) measurements were also conducted using a TRIAX 180 monochromator system with a 300 W xenon arc lamp light source and a Keithley 2410 semiconductor system at room temperature. We have modified the sentences in the revised manuscript (yellow color, line 115–117).

  • Line 215. How did you define the sensitivity? Is it similar to the paper: “Fabrication and Characterization of Ni-Doped ZnO Nanorod Arrays for UV Photodetector Application” or to the other paper “Fabrication and Characterization of UV Photodetectors with Cu-Doped ZnO Nanorod Arrays”? Is it the Iph/Idark ratio or is it [(Iphoto–Idark)/Idark] × 100 ? Please define explicitly the value of “sensitivity”.

Answer:

Thanks for your kind suggestion!

  • The sensitivity is the Iph/Idark ratio. We have modified the sentences in the revised manuscript (yellow color, line 214–215).

  • Again Line 215. How much is the dark current at 3V? Looking at Figure 6 a-b it looks like the dark current is approximately 2.5E-6 A for both the doped and undoped ZnO nanorods. So it would not be true that the Fe doping decreases the dark current. Moreover, the values I obtain for the sensitivity are 7 and 19 for the Iph/Idark definition, 600 and 1800 according to the [(Iphoto–Idark)/Idark] × 100 definition. In both cases I get very different values from the 43 and 475 that you report. Please write down all the dark and photo currents use to calculate the sensitivity. If you wish you can make a table similar to TABLE1 that you already made in “Fabrication and Characterization of UV Photodetectors with Cu-Doped ZnO Nanorod Arrays”.

Answer:

Thanks for your kind suggestion!

  • We have added the table 1 in the revised manuscript (yellow color, line 450–451 and 493–494), as seen in table 1.

  • Lines 215, 221 and 222. Reporting numbers with 4 or 5 digits means that you are assuming any error on the measurement to be smaller than 0.1% or even 0.01%, which I honestly doubt. Please provide proof of such small errors. Otherwise think again about the evaluation of uncertanties in your experiments and use only the appropriate number of digits.

Answer:

Thanks for your kind suggestion!

  • We have carried out experiments repeatedly and obtained similar results.

  • Line 222. In order to calculate the responsivity, you should have a calibrated measurement of the monochromated xenon lamp irradiance (W/m2). If so, can you add that information?

Answer:

Thanks for your kind suggestion!

  • A calibrated measurement of the monochromated xenon lamp irradiance was about 30 W/m2. We have added the sentences in the revised manuscript (yellow color, line 222–223).

Reviewer 2 Report

Manuscript reports on fabrication and testing ZnO and Fe-doped ZnO PDs by rf-sputtering. Reviewer found manuscript written clearly; however its novelty is limited. In general, the references cited demonstrated clearly that all major research aspects of this manuscript (e.g. Fe-doping to ZnO, UV photo detection and Oxygen sensing) were reported already. Authors failed to demonstrate by large extent on what is novelty of reported work, in other words, in what sense it is novel/better as compared to the literature and specific practical applications base on ZnO for UV detection.      

Authors shown that there has been an improvement in time responses due to Fe doping; however fundamental trade-off between the photocurrent gain and the speeds of PDs, which is of paramount importance for practical applications is missed. Also, current UV PDs figure-of-merit applicable to FZN shall be critically considered as a design guideline for the optimum Fe doping concentration for obtaining high performances in FZN-based PDs.

The discussion related to oxygen adsorption onto and desorption from the ZnO and FZN NR arrays are entirely based on the literature arguments without any experimental confirmation showing that the fabricated FZN NR-based MSM PDs have improved performance as compared to similar reported structures. Also, manuscript does not contain experimental ambient environment conditions in which PDs were tested, making reported results difficult to reproduce. Authors shall confirm the PD performance under controlled ambient oxygen/humidity/pressure conditions.  

Other issues:

What was the UV light spectral energy density used for testing PDs?

Fig.1 is rather simplistic and contains unnecessary graphical details (e.g. experimental systems). It is recommended that authors prepare more scientific like figure containing only pertaining details.

Was there any indication of Fe2O3 secondary phase admixture, there are seen new low intensity peaks in FZN, Fig.3? For example see here: DOI: 10.1155/2015/240948

Line 263, there is “Fermi energy”, shall be “Fermi energy”.

Fig.8 caption, there is “…and the electron-hole pairs were photogenerated under UV light emission”, and shall be”…and the electron-hole pairs were photogenerated under UV light irradiation.”

In conclusion, I do not recommend manuscript for publication without major revision.

Author Response

Reply to the comments of the editor and reviewer

Journal: Sensors (ISSN 1424-8220)

Manuscript Number: Sensors-853649

Title: Fabrication of Ultraviolet Photodetectors Based on Fe-Doped ZnO Nanorod Structures

=============================================================

Reviewer(s)' Comments to Author:

Reviewer: 2

Manuscript reports on fabrication and testing ZnO and Fe-doped ZnO PDs by rf-sputtering. Reviewer found manuscript written clearly; however its novelty is limited. In general, the references cited demonstrated clearly that all major research aspects of this manuscript (e.g. Fe-doping to ZnO, UV photo detection and Oxygen sensing) were reported already. Authors failed to demonstrate by large extent on what is novelty of reported work, in other words, in what sense it is novel/better as compared to the literature and specific practical applications base on ZnO for UV detection.

Authors shown that there has been an improvement in time responses due to Fe doping; however fundamental trade-off between the photocurrent gain and the speeds of PDs, which is of paramount importance for practical applications is missed. Also, current UV PDs figure-of-merit applicable to FZN shall be critically considered as a design guideline for the optimum Fe doping concentration for obtaining high performances in FZN-based PDs.

The discussion related to oxygen adsorption onto and desorption from the ZnO and FZN NR arrays are entirely based on the literature arguments without any experimental confirmation showing that the fabricated FZN NR-based MSM PDs have improved performance as compared to similar reported structures. Also, manuscript does not contain experimental ambient environment conditions in which PDs were tested, making reported results difficult to reproduce. Authors shall confirm the PD performance under controlled ambient oxygen/humidity/pressure conditions.

Answer:

Thanks for your kind suggestion!

  • We have perused the reviewers’ comments and suggestions, and made appropriate revisions in the revised manuscript to improve the clarity and delivery of the technical data. We also appreciate the reviewer for giving many constructive comments. By comparing with previous studies, we have added the table 2 in the revised manuscript (yellow color, line 269–273, 453–454, and 495–496), as seen in table 2 and reference 47-50.

  • What was the UV light spectral energy density used for testing PDs?

Answer:

Thanks for your kind suggestion!

  • The UV light spectral energy density (87 Jm-2 flash-1) is used for testing PDs below 380 nm.

  • 1 is rather simplistic and contains unnecessary graphical details (e.g. experimental systems). It is recommended that authors prepare more scientific like figure containing only pertaining details.

Answer:

Thanks for your kind suggestion!

  • We have modified the fabrication process of photodetector samples, as seen in Figure 1. (yellow color, line 456)

  • Was there any indication of Fe2O3 secondary phase admixture, there are seen new low intensity peaks in FZN, Fig.3? For example see here: DOI: 10.1155/2015/240948

Answer:

Thanks for your kind suggestion!

  • No reflection peaks of Fe2O3 (JCPDS Card No. 33-0664) or Fe3O4 (JCPDS Card No. 85-1436) can be seen in the diffraction patterns. It means that Fe atoms may substitute Zn2+ sites or incorporate interstitially in the ZnO structure, this phenomenon is similar to that of the earlier report for Fe-doped ZnO structure. See for example: “Noise Properties of Fe-ZnO Nanorod Ultraviolet Photodetectors”, IEEE Photonics Technology Letters, vol. 25, no. 21, November 1, 2013.

  • Line 263, there is “Fermi energy”, shall be “Fermi energy”.

Answer:

Thanks for your kind suggestion!

  • We have modified the sentences in the revised manuscript (yellow color, line 264).

  • 8 caption, there is “…and the electron-hole pairs were photogenerated under UV light emission”, and shall be”…and the electron-hole pairs were photogenerated in UV light irradiation.”

Answer:

Thanks for your kind suggestion!

  • We have modified the sentences in the revised manuscript (yellow color, line 447–448).

Reviewer 3 Report

  • In general, this is a reasonably well-written paper but there are some parts that need to be clarified.
  • The topic is timely and interesting.
  • My comments are focused on the electrical and optical characterization
  • The authors mention in several places that conductivity is improved by Fe doping. I do not doubt this, but can they quantify this statement?
  • If conductivity is increased, why is dark current reduced?
  • Put various parameters describing rise and fall characteristics (equations 12 and 13) into a table. I do not see the point of using (r1,r2) and (d1,d2) for comparison. Why not just use simple, extracted time constants?
  • Why are authors not using two different time constants, given that authors identify two different mechanisms for recombination: a) recombination at the deep defects, and b) electron trapping at the surface states. Please elaborate.
  • Why is Figure 6 b) so noisy while 6a) is clean?
  • Why are current levels in Fig. 6 a and b different than what is presented in Fig c) and d)? For example, from Fig. 6c) I would expect that the “on” current should be 2E-4 or 3E-4 (A) but Fig. 6a) saturates around 1.7E-5. Please explain.
  • This statement does not make sense – please clarify what you mean “Cyan block area under 380 nm UV illumination exhibits that the stabilities of pure ZnO and FZO samples are better.” What is being compared here?
  • Authors state that “The result displays all measured PDs from −3 V to +3 V bias voltages” but their plots are from -5V to + 5V.
  • State what the values of dark currents are.
  • Define sensitivity.
  • Explain how FZO affects Ag contacts and their Schottky barrier. It looks like the primary contact is between Ag and pure ZnO in the seed layer. I do not see how nanorods made of FZO can make the lateral contact to the Ag. Also, in another paper [45] the influence of oxygen and UV absorption on the barrier property was mentioned but for pure ZnO. How is this changed when FZO is used for nanorods? I think the authors should at least attempt to answer these questions.
  • Authors state “By contrast, FZO NR arrays with high surface volume ratio can lead to their adsorption onto and desorption from the surface of FZO [41-42]” where I assume that “contrast” is between FZO and Pure ZnO. But this does not make sense because both FZO and pure ZnO have “high surface volume ratio” so I do not see how this explains the better performance of FZO.
  • In general, I find the last paragraph before conclusions (lines 256 to 268) very general and confusing. I hope that authors can rewrite it and be specific about the mechanisms that improve FZO performance.

Author Response

Reply to the comments of the editor and reviewer

Journal: Sensors (ISSN 1424-8220)

Manuscript Number: Sensors-853649

Title: Fabrication of Ultraviolet Photodetectors Based on Fe-Doped ZnO Nanorod Structures

=============================================================

Reviewer(s)' Comments to Author:

Reviewer: 3

In general, this is a reasonably well-written paper but there are some parts that need to be clarified. The topic is timely and interesting. My comments are focused on the electrical and optical characterization. The authors mention in several places that conductivity is improved by Fe doping. I do not doubt this, but can they quantify this statement?

Answer:

Thanks for your kind suggestion!

  • We have perused the reviewers’ comments and suggestions, and made appropriate revisions in the revised manuscript to improve the clarity and delivery of the technical data. We also appreciate the reviewer for giving many constructive comments. This part is still in our investigation.

  • If conductivity is increased, why is dark current reduced?

Answer:

Thanks for your kind suggestion!

  • This reason is similar to that of the earlier report for Fe-doped ZnO structure. See for example: “Noise Properties of Fe-ZnO Nanorod Ultraviolet Photodetectors”, IEEE Photonics Technology Letters, vol. 25, no. 21, November 1, 2013.

  • Put various parameters describing rise and fall characteristics (equations 12 and 13) into a table. I do not see the point of using (r1,r2) and (d1,d2) for comparison. Why not just use simple, extracted time constants?

Answer:

Thanks for your kind suggestion!

  • We have added the table 1 in the revised manuscript (yellow color, line 450–451 and 493–494), as seen in table 1.

  • Why are authors not using two different time constants, given that authors identify two different mechanisms for recombination: a) recombination at the deep defects, and b) electron trapping at the surface states. Please elaborate.

Answer:

Thanks for your kind suggestion!

  • In this study, rise times (r1, r2) reveal the time for photocurrent rise to 50% and 90% of the peak value, respectively; and decay times (d1, d2) indicate the time for the photocurrent decays to 50% and 10% of the peak value, respectively. (yellow color, line 194–201). See for example: “Carrier relaxation through two-electron process during photoconduction in highly UV sensitive quasi-one-dimensional ZnO nanowires”, Phys. Lett., 2008, 93, 053102.

  • Why is Figure 6 b) so noisy while 6a) is clean?

Answer:

Thanks for your kind suggestion!

  • This part is still in our investigation.

  • Why are current levels in Fig. 6 a and b different than what is presented in Fig c) and d)? For example, from Fig. 6c) I would expect that the “on” current should be 2E-4 or 3E-4 (A) but Fig. 6a) saturates around 1.7E-5. Please explain.

Answer:

Thanks for your kind suggestion!

  • We have carried out repeated experiments and obtained similar results. Additionally, this phenomenon is similar to that of the earlier report for doping ZnO with different elements.

For example:(1) RSC Adv., 2014, 4, 31969–31972.

            (2) RSC Adv., 2018, 8, 32333–32343.

            (3) Materials Science and Engineering B 178 (2013) 1068–1075.

  • This statement does not make sense – please clarify what you mean “Cyan block area under 380 nm UV illumination exhibits that the stabilities of pure ZnO and FZO samples are better.” What is being compared here?

Answer:

Thanks for your kind suggestion!

  • As shown in the inset of Figure 6a–b, it exhibits that the reproducibility of pure ZnO and FZO samples is stability (Cyan block area is under 380 nm UV irradiation). We have modified the sentences in the revised manuscript (yellow color, line 207–209).

  • Authors state that “The result displays all measured PDs from −3 V to +3 V bias voltages” but their plots are from -5V to + 5V.

Answer:

Thanks for your kind suggestion!

  • The result displays all measured PDs from −5 V to +5 V bias voltages. We have modified the sentences in the revised manuscript (yellow color, line 212).

  • State what the values of dark currents are.

Answer:

Thanks for your kind suggestion!

  • We have added the table 1 in the revised manuscript (yellow color, line 450–451 and 493–494), as seen in table 1.

  • Define sensitivity.

Answer:

Thanks for your kind suggestion!

  • The sensitivity is the Iph/Idark ratio. We have modified the sentences in the revised manuscript (yellow color, line 214–215).

  • Explain how FZO affects Ag contacts and their Schottky barrier. It looks like the primary contact is between Ag and pure ZnO in the seed layer. I do not see how nanorods made of FZO can make the lateral contact to the Ag. Also, in another paper [45] the influence of oxygen and UV absorption on the barrier property was mentioned but for pure ZnO. How is this changed when FZO is used for nanorods? I think the authors should at least attempt to answer these questions.

Answer:

Thanks for your kind suggestion!

  • In general, Schottky forms are formed between metals and oxides. Meanwhile, the Fe is doped into ZnO structure, the conductivity will be enhanced.

  • Authors state “By contrast, FZO NR arrays with high surface volume ratio can lead to their adsorption onto and desorption from the surface of FZO [41-42]” where I assume that “contrast” is between FZO and Pure ZnO. But this does not make sense because both FZO and pure ZnO have “high surface volume ratio” so I do not see how this explains the better performance of FZO. See for example: Fe3+ center in ZnO. Rev. B, 1992, 45, 8997.

Answer:

Thanks for your kind suggestion!

  • After Fe-doping, the diameter and length of NRs were increased slightly, as seen in Figure 2. The results of this paper are similar to those of a previous report by Liu et al. and Sahai et al. (See for references 26–27). As a donor, Fe3+ ion usually provides a pair of electrons in the conduction band, thus improving the conductivity. See for example: Sensors and Actuators B 158 (2011) 9–16 and Journal of applied physics 116, 164315 (2014).

  • In general, I find the last paragraph before conclusions (lines 256 to 268) very general and confusing. I hope that authors can rewrite it and be specific about the mechanisms that improve FZO performance.

Answer:

Thanks for your kind suggestion!

  • I'm very sorry that we have tried our best to express the application of FZO PD.

Round 2

Reviewer 1 Report

Dear authors,

the modifications that you made clearly improved the quality of your paper.

Still I have a couple of minor points to address:

  • In table2, you added reference 48 including only the data of Pure ZnO nanorods. Please also add the values for the Fe-doped ZnO nanorods that you fabricated in 2013, so the improvement can be easily viewed.
  • In table2, you say that the Bias for the new Fe-doped nanorods is 3V, but in the text you clearly use two different voltages to obtain the parameters. In particular, the sensitivity is measured at 5V, while the responsivity is measured at 3V. This is quite confusing. Please either adjust all parameters to 3Volts, or make it clearer when you are using 3 vs 5 Volts

Author Response

Reply to the comments of the editor and reviewer

Journal: Sensors (ISSN 1424-8220)

Manuscript Number: Sensors-853649

Title: Fabrication of Ultraviolet Photodetectors Based on Fe-Doped ZnO Nanorod Structures

=============================================================

Reviewer(s)' Comments to Author:

Reviewer: 1

Dear authors, the modifications that you made clearly improved the quality of your paper. Still I have a couple of minor points to address:

In table 2, you added reference 48 including only the data of Pure ZnO nanorods. Please also add the values for the Fe-doped ZnO nanorods that you fabricated in 2013, so the improvement can be easily viewed.

Answer:

Thanks for your kind suggestion!

  • We have modified and added the sentences in the revised manuscript (yellow color, line 322), as shown in table 2.

In table 2, you say that the Bias for the new Fe-doped nanorods is 3 V, but in the text you clearly use two different voltages to obtain the parameters. In particular, the sensitivity is measured at 5 V, while the responsivity is measured at 3 V. This is quite confusing. Please either adjust all parameters to 3 Volts, or make it clearer when you are using 3 vs 5 Volts.

Answer:

Thanks for your kind suggestion!

  • -5 to +5 is just a range; the experimental results in this paper are mainly discussed in terms of 3 volts. We have modified and added the sentences in the revised manuscript (yellow color, line 242–244 and 252).

Reviewer 2 Report

Authors have addressed all reviewer’s concerns. Thank you! Manuscript can be accepted as is.

Author Response

Reply to the comments of the editor and reviewer

Journal: Sensors (ISSN 1424-8220)

Manuscript Number: Sensors-853649

Title: Fabrication of Ultraviolet Photodetectors Based on Fe-Doped ZnO Nanorod Structures

=============================================================

Reviewer(s)' Comments to Author:

Reviewer: 2

Authors have addressed all reviewer’s concerns. Thank you! Manuscript can be accepted as is.

Answer:

Dear reviewer 2

many many thanks!

YL Chu